# Selective Laser-Assisted Direct Synthesis of MoS_2_ for Graphene/MoS_2_ Schottky Junction

**DOI:** 10.3390/nano13222937

**Published:** 2023-11-13

**Authors:** Min Ji Jeon, Seok-Ki Hyeong, Hee Yoon Jang, Jihun Mun, Tae-Wook Kim, Sukang Bae, Seoung-Ki Lee

**Affiliations:** 1School of Material Science and Engineering, Pusan National University, Busan 46241, Republic of Korea; minji@pusan.ac.kr (M.J.J.);; 2Institute of Advanced Composite Materials, Korea Institute of Science and Technology, Wanju 55324, Republic of Korea; 3Advanced Instrumentation Institute, Korea Research Institute of Standards and Science, Daejeon 34113, Republic of Korea; 4Department of Flexible and Printable Electronics, Jeonbuk National University, Jeonju-si 54896, Republic of Korea; 5Department of JBNU-KIST Industry-Academia Convergence Research, Jeonbuk National University, Jeonju-si 54896, Republic of Korea

**Keywords:** selective laser annealing, graphene, molybdenum disulfide (MoS_2_), photothermal reaction, heterostructure, photodetector

## Abstract

Implementing a heterostructure by vertically stacking two-dimensional semiconductors is necessary for responding to various requirements in the future of semiconductor technology. However, the chemical-vapor deposition method, which is an existing two-dimensional (2D) material-processing method, inevitably causes heat damage to surrounding materials essential for functionality because of its high synthesis temperature. Therefore, the heterojunction of a 2D material that directly synthesized MoS_2_ on graphene using a laser-based photothermal reaction at room temperature was studied. The key to the photothermal-reaction mechanism is the difference in the photothermal absorption coefficients of the materials. The device in which graphene and MoS_2_ were vertically stacked using a laser-based photothermal reaction demonstrated its potential application as a photodetector that responds to light and its stability against cycling. The laser-based photothermal-reaction method for 2D materials will be further applied to various fields, such as transparent display electrodes, photodetectors, and solar cells, in the future.

## 1. Introduction

With the development of next-generation devices and the introduction of new functionalities, the primary objective for semiconductor technology is to increase integration density while achieving various form factors through downscaling and ultra-thinning techniques [1,2,3]. Pioneering research has highlighted the potential of thinning traditional materials such as silicon and oxide semiconductors to improve mechanical flexibility without compromising their quality; however, the innate three-dimensional (3D) crystalline structures of class materials imply inevitable limits to continuous downscaling strategies owing to several challenges such as increased surface energy, quantum effect-induced property changes, and thermal instability. In contrast, 2D crystalline materials, characterized by atomic interconnections in a 2D lattice, inherently retain their electrical, mechanical, and physical properties, even at the atomic scale [4,5]. For these reasons, they have attracted considerable attention as potential core materials for future electronic devices [6]. Notably, these 2D materials are highly compatible [7] with traditional Si-based complementary metal-oxide semiconductor (CMOS) manufacturing processes, leading to promising results in devices with hybrid structures [8,9]. 

Methods such as metal-organic chemical-vapor deposition (MOCVD), molecular-beam epitaxy, and atomic layer deposition have led to significant progress in the synthesis of transition-metal dichalcogenides, which are representative 2D semiconductors, on various substrates [10,11,12,13]. Even though these methods achieve high-quality materials capable of synthesizing single crystals or controlling the number of layers at the wafer scale, they rely on high synthesis temperatures (>700 °C) [14,15], post-thermal treatments, or specific substrates. Such dependence inevitably necessitates a transfer to the target substrate post-synthesis. Unfortunately, this transfer process typically results in unforeseen chemical contamination, physical wrinkles, and damage, making the prediction of the post-transfer quality and uniformity of the synthesized materials [16,17] increasingly challenging. To this end, laser-assisted synthesis methodologies have been proposed as potential solutions to overcome the limitations of conventional techniques [18,19,20,21,22]. Because of the monochromatic, coherent, and collimated nature of laser sources, photothermal reactions on materials can be systemically controlled by selectively [23,24,25,26] inducing lattice vibrations. This allows targeted thermal annealing even when the device chip comprises thermally fragile parts. In this context, the effectiveness of laser-assisted synthesis has been proven through research on the selective synthesis of 2D materials using lasers or the partial modulation of the interface properties of electronics [20,21,22]. Moreover, significant findings are emerging, demonstrating the direct synthesis of 2D heterostructures beyond the scope of individual materials [27,28,29,30,31]: This is accomplished by integrating distinct 2D materials, particularly n-type MoS_2_ with p-type WS_2_, eliminating the need for a transfer process. Therefore, various heterojunction structures based on 2D materials need to be implemented and validated by exploiting the universal applicability of selective-laser photothermal reactions.

Herein, we implement a graphene/MoS_2_ heterojunction structure by integrating n-type MoS_2_ on semi-metallic graphene via a laser-assisted selective photothermal-reaction method by optimizing the photothermal and laser parameters. For the synthesis of MoS_2_, a thermally decomposable (NH_4_)_2_MoS_4_ precursor was mixed with an organic solvent, and a fiber laser (λ=1.06 μm) with a high absorption rate for the MoS_2_ precursor was selected to prevent damage to the underlying graphene and substrate. Unlike the traditional CVD synthesis method, controlling the thermal-spreading range using the laser parameters suppressed damage to the surrounding layers, such as the underlying graphene and SiO_2_/Si wafer, which provided a means to implement a graphene/MoS_2_-based photodetector with a robust interface. 

## 2. Experimental Section

### 2.1. Materials

Poly(methylmethacrylate) ((C_5_O_2_H_8_)_n_; PMMA), ammonium tetrathiomolybdate((NH_4_)_2_MoS_4_), CAS No. 15060-55-6), and hexamethyldisilazane((CH_3_)_3_SiNHSi(CH_3_)_3_, CAS No. 999-97-3) were purchased from Sigma-Aldrich (St. Louis, MI, USA). A photoresist (AZ 5214-E; PR, LOT NO. USAW417063) and Developer (AZ 300 MIF developer, LOT NO. KR387425) were purchased from Merck (Rahway, NJ, USA). Dimethyl sulfoxide ((CH_3_)_2_SO, CAS No. 67-68-5) and ammonium persulfate((NH_4_)_2_S_2_O_8_, CAS No. 7727-54-0) were purchased from Daejung Chemicals (Siheung, Republic of Korea).

### 2.2. Graphene Synthesis and Transfer

Monolayer graphene was grown via CVD on 25 µm-thick copper foils [32] at temperatures close to 1000 °C using CH_4_ (125 sccm) as a carbon source and H_2_ (100 sccm) as a reactant gas [33]. The PMMA solution was spin-coated onto the graphene at 3000 rpm for 30 s to serve as a supporting layer. the specimen was annealed on a hot plate at 100 °C for 1 min. The surface of the graphene/copper foils was treated with an O_2_ plasma process (power 150 W, time 30 s) to improve the copper foils’ wettability. Subsequently, graphene was separated by etching the Cu foil in an ammonium persulfate solution (2 g ammonium persulfate in 100 mL deionized (DI) water) for 4 h. Once all of the copper was etched away, only the PMMA/graphene film remained, which was lifted using a glass slide and rinsed twice in DI water. The PMMA/graphene layer was then transferred to a 300 nm SiO_2_/P+ Si substrate. Finally, to remove PMMA from the graphene, it was immersed in acetone in two steps: first for 30 min and then for an additional 90 min.

### 2.3. Graphene Patterning, Thermal Cleaning, and Doping

The formation of the graphene-patterned array was achieved using typical photolithography and oxygen plasma etching methods. The graphene was uniformly covered by the PR solution, spinning at 3000 rpm for 30 s, and then baked at 95 °C for 2 min. Subsequently, the sample was exposed to aligner UV light for 15 s, creating a graphene-pattern array with a size of 100 μm × 300 μm. The samples were rinsed with a developer, followed by a rinse with deionized water, and then dried using nitrogen. The exposed graphene regions, not protected by PR, were etched through a reactive ion etching process. The sample was then soaked in acetone for about 30 min to remove the PR from the graphene. As a result, a graphene-pattern array was obtained. To thoroughly remove the PMMA and PR residues that remained after the acetone treatment, the patterned graphene underwent hydrogen annealing (H_2_) at 300 °C for 20 min, ensuring a clean surface [34,35,36]. Then, the patterned graphene was doped for 12 h to be completely submerged in a hexamethyldisilazane (HMDS) solution. Additionally, the doped specimen was dried with nitrogen gas. 

### 2.4. MoS_2_ Synthesis on Graphene

To create a vertically stacked MoS_2_ heterostructure on patterned graphene, a 0.633 M solution was formulated by mixing 1.0 g of (NH_4_)_2_MoS_4_ precursor with 6 mL of dimethyl sulfoxide (DMSO). A multistep coating technique ensured uniform coverage of the precursor solution on the graphene. SiO_2_/Si wafers (with a 300-nm-thick SiO_2_ layer) underwent two spin-coating stages: an initial 10 s spin at 500 rpm, followed by a 30 s spin at 2500 rpm. Subsequently, the specimen was annealed on a hot plate at 150 °C for 3 min to evaporate any remaining solvent. A fiber laser with a 1.06 μm wavelength and 20 W output was employed to selectively anneal the precursor layer, thereby synthesizing MoS_2_. The parameters for optimizing MoS_2_ synthesis via the laser were as follows: (i) laser power (from 1 W to 20 W), (ii) scan speed (from 10 mm/s to 500 mm/s), and (iii) frequency (from 20 kHz to 200 kHz). After the selective synthesis of MoS_2_, the untreated parts were washed away by immersion in DMSO solvent.

### 2.5. Device Characterization

The I–V characteristics of the device were measured under low-vacuum conditions (5 × 10^−3^ Torr) using a probe station connected to a parameter analyzer (4200, Keithley, Cleveland, OH, USA). To confirm the optoelectronic characteristics, the devices were illuminated using a white-light halogen lamp, and the light intensities were measured using a power meter (PM 100d, Thorlabs, Newton, NJ, USA).

## 3. Results and Discussion

Figure 1a,b illustrates a schematic of the graphene/MoS_2_-heterostructure fabrication through laser-based photothermal synthesis, complemented by the corresponding optical-microscopy images for each step. Initially, graphene was synthesized using the conventional CVD method [33,37], which enabled the uniform production of monolayer graphene over a large area. The graphene was then transferred to a SiO_2_/Si wafer using a PMMA supporting layer. Following the removal of the supporting layer, a rectangular graphene pattern (100 μm × 300 μm) was defined using photolithography and reactive ion etching with O_2_ plasma [38]. During these transfer and patterning stages, water molecules may be trapped between the graphene and substrate, or contamination from organic compounds may occur. Therefore, the patterned graphene was annealed via a thermal-cleaning process at 300 °C in a hydrogen atmosphere for 20 min. Furthermore, a self-assembled monolayer (SAM) of HMDS was applied as a buffer layer to the graphene via a dipping process. This layer mitigated unintended doping effects on the graphene and protected the graphene from the subsequent laser-synthesis step. Subsequently, an ammonium tetrathiomolybdate ((NH_4_)_2_MoS_4_; ATM) precursor solution was spin-coated onto the patterned graphene. Considering the contrasting nature of the graphene pattern (hydrophobic) and SiO_2_ surface (hydrophilic), achieving a uniform precursor coating on this nonuniform surface is challenging. The optimization strategies for these coatings are discussed in Figure 2. After forming a uniform precursor film, we employed a fiber laser (λ = 1.06 µm) to synthesize the MoS_2_ pattern on the preformed graphene pattern. We chose a fiber laser among the laser sources with various wavelengths because of its low optical absorption in the surrounding materials, such as graphene, SiO_2_, and Si; however, it has high optical absorption only in the (NH_4_)_2_MoS_2_ precursor. Therefore, only the precursor layer could be selectively heated and thermally decomposed into MoS_2_ [39]. In addition, the fiber laser had a low absorption rate in MoS_2_, even if MoS_2_ was synthesized after the precursor was thermally decomposed; therefore, the laser penetrated the MoS_2_ [23] without thermal damage. This was the key principle for realizing the graphene/MoS_2_ heterostructure by synthesizing MoS_2_ without damaging graphene in the lower layer. More importantly, this direct-synthesis method can realize a vertically stacked layer structure without using a conventional transfer method, thereby fundamentally avoiding problems such as wrinkles or contamination traps at the interface. The optimized laser conditions on the SiO_2_/Si substrate were as follows: a laser power of 10.8 W, scan speed of 100 mm/s, and frequency of 20 kHz. Finally, the source and drain were affixed using shadow masks to analyze the electrical properties of the graphene/MoS_2_ heterojunction. Figure 1c shows Raman spectra obtained from each region of the material at an excitation wavelength of 532 nm. The G and 2D peaks were observed at ~1599 cm^−1^ and ~2700 cm^−1^, respectively (marked in red). The 2D peak exhibited a sharp, single Lorentzian line shape and was considerably more intense than the G peak, which is a characteristic feature of monolayer graphene. Additionally, the D peak (~1360 cm^−1^), which indicates defects and imperfections within the graphene crystal, was barely discernible. The Raman spectra of MoS_2_ synthesized via the laser-based photothermal reaction are shown in blue. Two prominent MoS_2_ peaks, E_2g_ and A_1g_, could be distinctly observed at 383 cm^−1^ and 409 cm^−1^, respectively. A Raman characteristic of MoS_2_ is the narrowing of the gap between these two peaks as it transitions from the bulk state to that of a single layer [40,41]. This allowed the determination of the number of MoS_2_ layers present. The gap between these peaks was approximately 26 cm^−1^, confirming the few-layered nature of MoS_2_. Atomic force microscopy (AFM) analysis has substantiated that the synthesized molybdenum disulfide (MoS_2_) exhibits a consistent and uniform film-like structure with a measured thickness of 10 nm. The thickness control of the MoS_2_ film can be achieved through the regulation of solvent concentration and the adjustment of spin-coating velocity. The critical aspect to consider is that to achieve uniform synthesis of MoS_2_ in film form, it is necessary to adjust the laser scanning speed in accordance with changes in the thickness of the precursor film (Appendix A). Specifically, thicker precursor layers demand a greater amount of photothermal energy for complete thermal decomposition into MoS_2_, thereby necessitating a reduction in the laser scanning speed. According to previous research results that analyzed the surface of MoS_2_ synthesized using this method, the MoS_2_ yields a root mean square (RMS) surface roughness of 1.15 nm [23]. Interestingly, these major peaks were also maintained in the heterojunction where graphene and Mos_2_ overlapped (green dot in Figure 1b), as evidenced by the green colored Raman spectra. The strength of the D peak was also similar to that of graphene at its initial intensity. For comparison, we prepared a graphene/MoS_2_ control sample in which MoS_2_ was thermally decomposed from the (NH_4_)_2_MoS_4_ precursor using thermal CVD, and we analyzed its Raman characteristics. A comparison between the Raman peak of the graphene/MoS_2_ structure synthesized by laser irradiation (marked in green) and that of the graphene/MoS_2_ structure created by thermal chemical vapor deposition (T-CVD) (marked in black) showed that the Raman characteristics of MoS_2_ were similar, whereas those for graphene exhibited a pronounced difference. When MoS_2_ was directly synthesized using general T-CVD, the underlying graphene sustained thermal damage, leading to an amplified D peak and diminished 2D peak. Therefore, our research demonstrates that the laser-based selective thermal-treatment method can be used to directly synthesize MoS_2_ layers on a graphene monolayer without causing damage. This mechanism, when paired with other 2D materials that are amenable to thermal decomposition, holds promise for the assembly of diverse heterojunction structures.

The uniform deposition of the (NH_4_)_2_MoS_4_ precursor layer on graphene is essential to demonstrate a MoS_2_/graphene-heterojunction structure. In addition, a uniform and continuous pinhole-free thin film with a large area enabled the fabrication of a device array with a high yield. However, because graphene has a hydrophobic-surface characteristic owing to its unique atomic-crystal [42] structure, a difference was observed in the surface homogeneity between the spin coating of the ATM solution on the SiO_2_ substrate and the graphene-transferred SiO_2_ substrate, as shown in Figure 2a. To enhance the surface uniformity of the ATM film on graphene, three different solvents—dimethylformamide (DMF) mixed with 2-aminoethanol, a Triton X-100 mixture, and DMSO—were studied. In the case of the SiO_2_/Si substrate, the surface of the wafer was treated with an O_2_ plasma process (power 150 W, time 30 s) prior to spin-coating the ATM solution to improve the surface wettability. As shown in the optical-microscopy images in Figure 2b–d, a monotonous and uniform color indicated that the ATM solvent was evenly distributed in all types of samples. However, plasma treatment is not preferred for graphene surfaces because activated oxygen ions can severely damage the graphene, degrading its electrical performance. Therefore, we attempted to overcome this problem by adjusting the interactions between graphene and the solvent. In dissolved ATM solutions, balancing the viscosity and surface tension of the precursor is important. Initially, DMF and 2-aminoethanol were employed as the solvent and additive [43], respectively. DMF has been used as a common organic solvent to dissolve ATM precursors [39], and amine group-based additive molecules have been used to stabilize ionic thiomolybdate (MoS_4_^2−^) clusters. However, as a result of numerous mixing control experiments between solutions, when DMF + 2-aminoethanol-based solvent was used, obtaining stable coating conditions was difficult as most of the graphene layers rolled after peeling from the substrate as indicated by the arrow in Figure 2e. Although we attempted to use another amine-based additive (*n*-butylamine), a similar phenomenon occurred when the solvent penetrated the substrate and graphene, resulting in delamination. As an alternative, the surface tension of the solution was controlled by adding Triton X-100 surfactant to the ATM solution. As a non-ionic surfactant, the hydrophobic group of Triton-X-100 engaged in interactions with the carbonaceous surface, leading to its adsorption onto the graphene surface. This adsorptive phenomenon enhanced the wettability of the graphene substrate, thereby facilitating a more uniform dispersion of the solvents on its surface [44]. Figure 2f appears to show improved homogeneity; however, complete homogeneity across the graphene surface was not achieved. As a result, the graphene was partially peeled off from the substrate as indicated by arrow. Interestingly, although DMSO is a polar organic solvent similar to DMF, experimental observations indicated that the ATM solution uniformly coated the graphene without causing any damage when DMSO was used as the solvent [45]. Based on previous experimental procedures, it is speculated that the balance between the surface tension of DMSO, the ionic thiomolybdate cluster, and the van der Waals forces [46] between the graphene and SiO_2_ interface may allow for an even coating of the ATM precursor without peeling off the graphene (Figure 2g). Finally, when the ATM solution was spin-coated onto the SiO_2_ substrate, it was confirmed that the DMF, Triton X-100 mixture, and DMSO were homogeneously coated on the surface. However, to spin-coat the graphene-transferred SiO_2_ substrate, it was confirmed that the surface could be coated homogeneously using the DMSO solvent.

The electrical-junction properties of the fabricated graphene/MoS_2_ were investigated. First, an HMDS layer was deposited on the graphene surface to minimize the unintentional doping effect caused by the Cu etchant and supporting polymer during the transfer process of the CVD graphene. That is, the hydrophobic SAM including HMDS suppressed the charge-impurity scattering effect caused by the impurity and simultaneously improved overall uniformity [47,48]. In addition, the HMDS buffer layer prevented damage to graphene from additional processes such as metal deposition [49] and laser annealing. Figure 3a illustrates the HMDS-doped graphene-based back-gate transistor on the Si/SiO_2_ substrate. The HMDS treatment was conducted for 12 h on the graphene transferred onto the SiO_2_/Si substrate prior to the deposition of the contact electrodes. The Raman spectra of graphene before and after HMDS doping are shown in Figure 3b. The Raman-frequency values of the G and 2D modes were sensitive to charge impurities and strains in graphene, enabling us to confirm the influence of HMDS. The G and 2D peak positions of pure graphene were 1600 cm^−1^ and 2695 cm^−1^, respectively (marked in black); however, after HMDS formation, both peaks shifted down to 1595 cm^−1^ and 2686 cm^−1^, respectively (marked in red). The Raman shift indicated that although the surface of graphene had hydrophobic characteristics, the entire surface was covered well with HMDS because some existing defects or oxidized species in graphene played a role in nucleating the SAM [49]. Moreover, the extracted value of Δ2D/ΔG was less than 2 [50], which indicates that no effective stress was applied to graphene after HMDS deposition. A comparative assessment of the Fermi energy and carrier mobility of pristine and HMDS-doped graphene is shown in Figure 3c. For the bare graphene, a gate voltage (Dirac point) was observed at 33.1 V. However, after the HMDS-doping process, a pronounced Dirac-point shift to lower gate voltages was repeatedly observed (V_G_: 22.2 V) [48,51,52,53]. Based on the electronic structure of graphene, the Dirac-point shift may be associated with the change in Fermi energy. Because of the linear band structure of graphene, the shift in Fermi energy caused by the HMDS dipole can be described as follows [54]:$$\Delta E=\hbar v_{F}\sqrt{\pi n\left(V_{2}-V_{1}\right)},$$
where *ν_F_* is the Fermi velocity previously reported as 1.1 × 10^6^ m/s, *n* is the intrinsic carrier density per volt with a typical value of approximately 7.2 × 10^10^ cm^−2^ V^−1^, and *V*_1_ (*V*_2_) is the Dirac point of devices without (with) HMDS treatment. The calculation confirmed that the HMDS treatment shifted the Fermi-energy level of graphene by 0.11 eV closer to the vacuum level. The field-effect carrier mobility of graphene was additionally calculated from the maximum slope near the charge-neutrality point, as follows:$$\mu_{FET}=\frac{L_{ch}g_{m}}{W_{ch}C_{ox}V_{D}}$$
where *L_ch_* and *W_ch_* are the channel length and width, respectively, *g_m_* is *dI_D_*/*dV_G_*, *C_ox_* is the gate oxide capacitance per area, and *V_D_* is the applied drain-source voltage. As plotted in Figure 3c, the hole and electron carrier mobilities of HMDS-treated graphene showed values of ~450.8 cm^2^/Vs and ~297.9 cm^2^/Vs, respectively. This represents a significant enhancement in mobility compared to bare graphene (μ_hole_: ~86.5 cm^2^/Vs and μ_elec_: ~95.4 cm^2^/Vs). The electrical properties of MoS_2_, interfaced with graphene, were also analyzed via transfer characteristics (Appendix A). The MoS_2_ synthesized via laser based photothermal processing demonstrated comparatively lower electron mobility, ranging from 1 to 5 cm^2^/Vs, than that synthesized through chemical vapor deposition (CVD) methods, with an observed on/off ratio of 10^3^. These moderate properties can be attributed to the fine-grained structure of the laser-synthesized MoS_2_. Determining the electronic band structure of the synthesized graphene/MoS_2_ heterostructure is necessary for analyzing its electrical characteristics and understanding its charge-transfer dynamics. The work function of each material was experimentally measured using ultraviolet photoelectron spectroscopy. The work function was calculated as follows [33]: Φ = *hν* − (*E_SECO_* − *E_VB_*),
where *hν* is the energy of the incident UV photons (He I line = 21.21 eV), *E_SECO_* is the secondary electron-cutoff energy, and *E_VB_* is the onset of the valence band. As expected, bare graphene exhibited significant fluctuations in the analysis value, ranging from 4.3 eV to 5.2 eV depending on the measurement position, whereas HMDS-doped graphene exhibited a stable value of approximately 4.5 eV. Furthermore, MoS_2_ synthesized by laser treatment exhibited a uniform work function of approximately 4.2 eV, which was in good agreement with that of the few-layered MoS_2_ [21]. Figure 3f illustrates the band alignment of the graphene/MoS_2_ heterojunction, delineated according to the positions of the respective band structures. In a thermal-equilibrium state, the band structure bends at the interface due to the difference in work functions between graphene and MoS_2_, creating a Schottky barrier (Φ*_B_*) and built-in potential (*V_bi_*) on the semiconductor side. For an applied external bias, the Fermi level of graphene shifted downward during the application of a reverse bias owing to the decrease in negative charges moving toward the graphene, whereas the Fermi level of the n-type MoS_2_ shifted upward, as illustrated in Figure 3g. This upward shift of the MoS_2_ Fermi level relieved the band bending, leading to a decrease in the built-in potential. Conversely, when a forward bias was applied to the graphene/MoS_2_ Schottky junction, the Fermi level of graphene shifted upward, whereas that of the MoS_2_ shifted downward. This occurred because more positive charges were localized in the MoS_2_ and more negative charges were transported to the graphene, as depicted in Figure 3h. Unlike the scenario under reverse bias, the value of the built-in potential increased during the application of a forward bias. When a reverse bias was applied, the majority carriers of MoS_2_ and graphene easily overcame the lowered built-in potential and transferred to the counter materials beyond the junction. However, in the case of a forward bias, the barrier at the junction was high; therefore, the charges were greatly restricted when moving toward each other. Therefore, owing to the aforementioned charge-transfer dynamics, the graphene/MoS_2_ junction, which was MoS_2_ directly synthesized on graphene using a laser-assisted methodology, exhibited rectification characteristics that depended on the applied forward-to-reverse bias. However, when MoS_2_ was grown via the thermal CVD process on the graphene, the underlying graphene experienced pronounced thermal degradation. These harsh synthesis conditions led to a substantial decline in conductivity, or in certain instances, a failure to establish a Schottky barrier, thereby inhibiting any rectifying behavior, as shown in Figure 3e. 

The developed graphene/MoS_2_-based Schottky junction was characterized in terms of its optical responsiveness to external light by periodically turning the light on and off. To this end, the optoelectronic device employs the configuration mentioned in Figure 1b, wherein the intersection of the graphene and MoS_2_ patterns spans an area of 100 µm × 100 µm. Detailed specifications of the device dimensions are described in the Appendix A, accompanied by illustrative schematics. We used a commercial visible-light halogen lamp (FOK-100 W, Fiber Optic Korea Co., Ltd., Cheonan, Republic of Korea) with a peak wavelength of 650 nm as the light source. As shown in Figure 4a–d, a difference in photocurrent was observed depending on the bias strength even under a consistent light intensity of 1.92 mW/cm^2^. To clarify, the optoelectronic device we have constructed incorporates two Schottky junctions encompassing both the graphene/MoS_2_ junction and MoS_2_/metal electrode junction. We assessed the photocurrent induced in each region utilizing a focused monochromatic laser beam (refer to Appendix A). The result of Appendix A confirmed that the photocurrent generated at the MoS_2_/metal electrode junction was markedly lower than that at the graphene/MoS_2_ junction. The dominant mechanisms that influenced the photocurrent in graphene/MoS_2_-based optoelectronic devices were photoconductivity and photogating [55]. The difference in photoconductivity appeared as an increase in the number of free charge carriers owing to photon absorption. Additionally, charge trapping occurred owing to disorders and defects, resulting in differences in photogating. We plotted the change in photocurrent upon light activation against the photocurrent variation in darkness (*I*/*I*_0_), testing two distinct bias voltages under forward and reverse biases. A larger photocurrent is observed in the case of forward bias. This can be considered to be due to the effect of the internal barrier depending on the direction of the applied bias as mentioned in Figure 3.

When considering the rise time (*τ*_rise_) and decay time (*τ*_decay_) of the charge carrier, which represent the duration of charge transfer and production of the maximum output current (ranging between 10 and 90% intensity), respectively, we extracted the following values: *τ*_rise_ = 10.3 s (*τ*_decay_ = 8.49 s) at −0.7 V, 10.96 s (8.9 s) at −0.2 V, 4.6 s (7.26 s) at 0.2 V, and 6.41 s (7.74 s) at 0.7V [56,57]. From an on/off photoresponsivity-ratio perspective, the reverse bias exhibited enhanced light reactivity. However, the forward bias demonstrated swifter responsiveness. For a comparative analysis, we fabricated graphene/MoS_2_-based junction devices using the CVD method and evaluated their characteristics. Upon analyzing the photoresponse under the same illumination conditions, it was found that the device fabricated using CVD synthesis exhibited negligible photocurrent generation. A detailed examination of the individual components revealed that the graphene had almost entirely lost its conductivity. These findings suggest that the graphene underwent significant damage during the MoS_2_ synthesis process (process temperature > 700 °C), which accounts for the absence of the rectifying effect as presented in Figure 3e. The tendency of currents to respond to light was shown to increase monotonically and proportionally as the intensity of the light increased linearly with no evidence of hysterysis (Figure 4e). We also investigated the temporal photoresponse, which indicated the potential of a photodetector, by monitoring the photocurrent under pulsed illumination in the frequency range of 0.05–1 Hz (Figure 4f). The graphene/MoS_2_-based Schottky diode was highly responsive to the incident pulsed light, with excellent stability and reproducibility over multiple cycles.

## 4. Conclusions

In this study, we introduced a facile approach for the selective synthesis of MoS_2_ on graphene by employing laser-based photothermal treatment, which enabled the direct formation of graphene/MoS_2_ heterostructures. Notably, this innovative technique offered a means of designing heterostructures from thermally sensitive materials by confining the heat treatment to the intended layer, effectively addressing the challenges that existing T-CVD methodologies cannot overcome. Central to our study is the understanding that the choice of the laser source is crucial. It must efficiently induce the thermal decomposition of the precursor without compromising the structural integrity of the synthesized MoS_2_ and graphene. This balance was achieved by harnessing the differential light-absorption rates of the ATM precursor and MoS_2_. Furthermore, we provided solutions for mitigating the surface irregularities introduced by graphene, thereby paving the way for achieving a uniform coating. The introduction of an HMDS buffer layer was a pivotal step that safeguarded the electrical characteristics of graphene throughout the fabrication sequence. Comprehensive evaluation of the electrical properties of the synthesized graphene/MoS_2_ heterostructures yielded encouraging results. Although this study did not specifically address contact resistance issues, it expects that the optimization of contact resistance between the two-dimensional semiconductor and the metal electrode could significantly enhance the operational efficiency of the photoelectric device [58,59,60,61,62,63]. As we venture deeper into the expansive field of this pioneering synthesis technique, we anticipate its widespread adoption and its pivotal role in driving next-generation devices and technologies.

## Figures and Tables

**Figure 1 nanomaterials-13-02937-f001:**
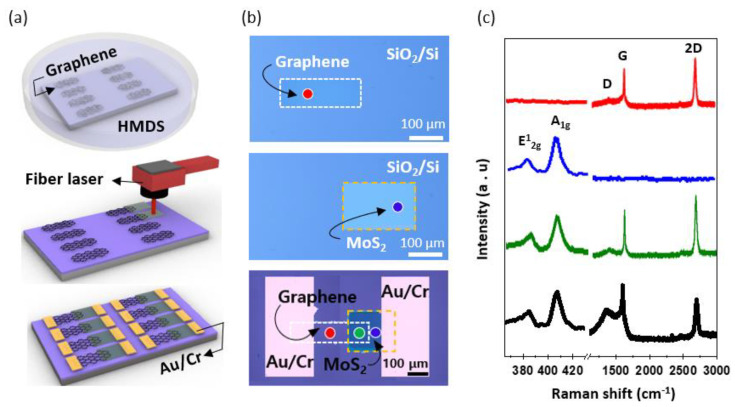
(**a**) Schematic diagram of the device manufacturing process, showing the stacking of MoS_2_ on graphene; (**b**) OM image of patterned graphene, MoS_2_, and a device made by a graphene/MoS_2_ heterostructure; (**c**) Raman spectra for CVD graphene (red), MoS_2_ synthesized with a laser (blue), vertically stacked MoS_2_ (synthesized with a laser) on graphene (green), and vertically stacked MoS_2_ (synthesized with CVD) on graphene (black).

**Figure 2 nanomaterials-13-02937-f002:**
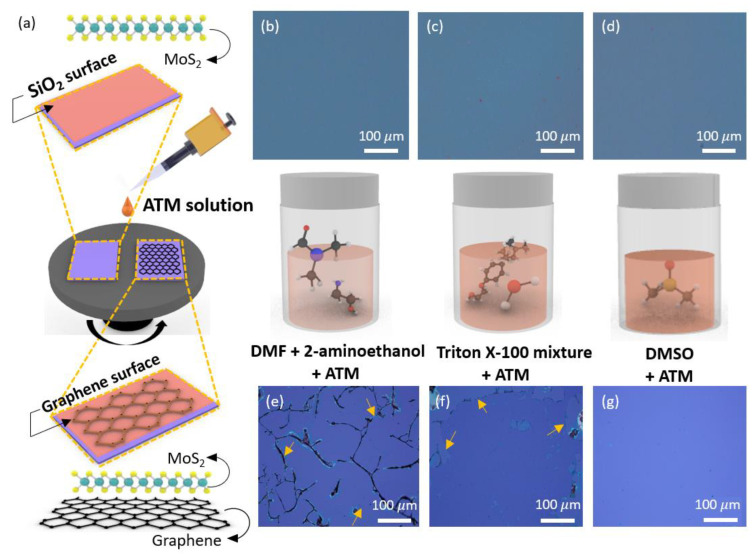
(**a**) Schematic illustration of the coating process for the (NH_4_)_2_MoS_4_ precursor on SiO_2_ and graphene/SiO_2_ surfaces. Optical-microscopy images showing the (NH_4_)_2_MoS_4_ precursor spin-coated onto a SiO_2_ surface with different solvents: (**b**) DMF + 2-aminoethanol, (**c**) Triton X-100 mixture, and (**d**) DMSO. Optical-microscopy images of the (NH_4_)_2_MoS_4_ precursor spin-coated onto a graphene/SiO_2_ substrate using the following solvents: (**e**) DMF + 2-aminoethanol, (**f**) Triton X-100 mixture, and (**g**) DMSO.

**Figure 3 nanomaterials-13-02937-f003:**
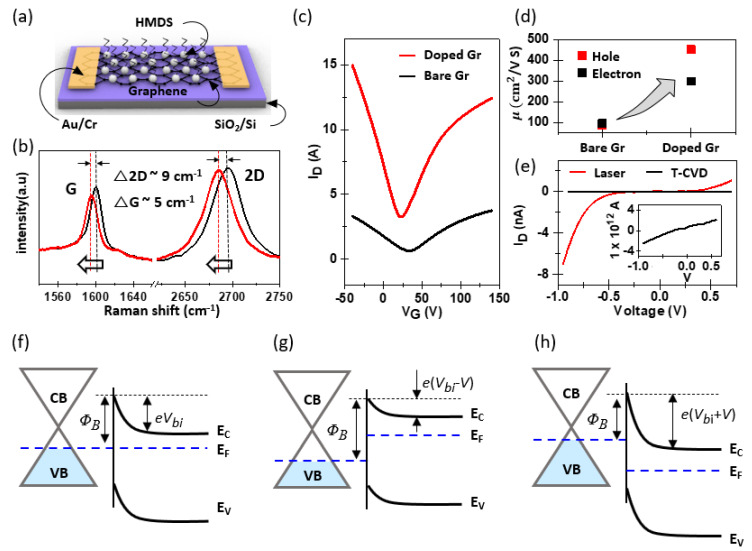
(**a**) Three-dimensional schematic diagram of an HMDS-doped graphene transistor; (**b**) Raman peak shift of graphene before and after HMDS doping; (**c**) transfer characteristic of graphene transistor with and without HMDS; (**d**) comparison of hole and electron carrier mobility before and after HMDS doping; (**e**) comparison of the rectification characteristics between graphene/MoS_2_ produced by the laser method and graphene/MoS_2_ produced by the T-CVD method. (Inset: detailed current characteristics of the device fabricated by T-CVD: (**f**) band diagram of graphene/MoS_2_ heterostructure; (**g**) reverse-bias condition; (**h**) forward-bias condition.

**Figure 4 nanomaterials-13-02937-f004:**
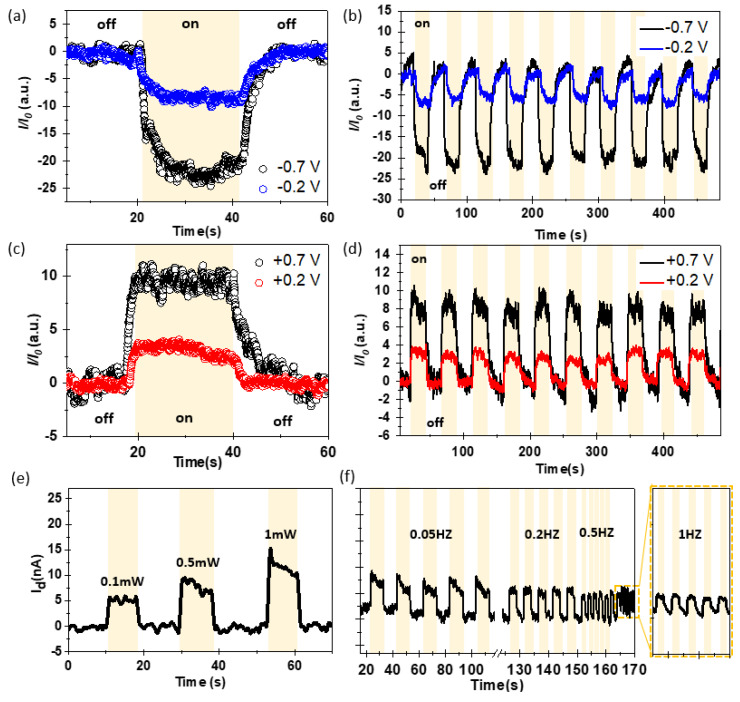
(**a**) Photocurrent characteristic of graphene/MoS_2_ heterostructure under reverse bias; (**b**) cycle stability under reverse bias; (**c**) photocurrent characteristic of graphene/MoS_2_ heterostructure under forward bias; (**d**) cycle stability under forward bias; (**e**) current–time characteristic with power of light source; (**f**) current–time characteristic with frequency.

## Data Availability

Data are contained within the article and Appendix A.

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
