# Peer review of "Selective Laser-Assisted Direct Synthesis of MoS2 for Graphene/MoS2 Schottky Junction"

_nanomaterials, 2023, doi:10.3390/nano13222937_

Round 1

Reviewer 1 Report

Comments and Suggestions for Authors

Comments on the Quality of English Language

Author Response

Thank you for your valuable comments so that we can improve the quality of the submitted manuscript.

We have prepared a point-by-point response to your comments.

Thank you for your time.

Seoung-Ki Lee

Reviewer 2 Report

Comments and Suggestions for Authors

In this manuscript, the authors reported the laser-assisted direct synthesis of MoS2 on graphene to form the graphene/MoS2 heterojunction. The authors investigated the optimal condition to facilitate the direct synthesis of MoS2 on patterned graphene strip. The as-prepared graphene/MoS2 heterojunction shows good photo-responsive behavior. I recommend the acceptance after the following concerns are well addressed.

1. It seems the synthesis of MoS2 is highly dependent on the laser. However, the influence of the amount of ATM on the laser power and scanning speed is not discussed in detail. If ATM with high concentration is used, what happens? If thicker MoS2 film is deposited on graphene, does the device performance become better or not?

2. The authors show provide the morphology of MoS2 after thermal decomposition by laser scanning. The ATM is converted to MoS2 flake or nanoparticle? AFM and SEM images should be useful to confirm it.

3. Compare to the laser-assisted graphene/MoS2, the device performance of stacked CVD-MoS2/graphene heterojunction is better or comparable?

4. In the inset of Figure 3e, it should be 10^-12, not 10^12.

Comments on the Quality of English Language

 Minor editing of English language required

Author Response

(The authors gave the same response as above.)

Round 2

Reviewer 2 Report

Comments and Suggestions for Authors

I think the authors have addressed my concerns and recommend acceptance in its current form.

Comments on the Quality of English Language

Minor editing of English language required